# A DFT Study of Ruthenium *fcc* Nano-Dots: Size-Dependent Induced Magnetic Moments

**DOI:** 10.3390/nano13061118

**Published:** 2023-03-21

**Authors:** Marietjie J. Ungerer, Nora H. de Leeuw

**Affiliations:** 1School of Chemistry, Cardiff University, Cardiff CF10 3AT, UK; 2School of Chemistry, University of Leeds, Leeds LS2 9JT, UK

**Keywords:** DFT, ruthenium, Ru, *fcc*, nano-dots, magnetisation

## Abstract

Many areas of electronics, engineering and manufacturing rely on ferromagnetic materials, including iron, nickel and cobalt. Very few other materials have an innate magnetic moment rather than induced magnetic properties, which are more common. However, in a previous study of ruthenium nanoparticles, the smallest nano-dots showed significant magnetic moments. Furthermore, ruthenium nanoparticles with a face-centred cubic (*fcc*) packing structure exhibit high catalytic activity towards several reactions and such catalysts are of special interest for the electrocatalytic production of hydrogen. Previous calculations have shown that the energy per atom resembles that of the bulk energy per atom when the surface-to-bulk ratio < 1, but in its smallest form, nano-dots exhibit a range of other properties. Therefore, in this study, we have carried out calculations based on the density functional theory (DFT) with long-range dispersion corrections DFT-D3 and DFT-D3-(BJ) to systematically investigate the magnetic moments of two different morphologies and various sizes of Ru nano-dots in the *fcc* phase. To confirm the results obtained by the plane-wave DFT methodologies, additional atom-centred DFT calculations were carried out on the smallest nano-dots to establish accurate spin-splitting energetics. Surprisingly, we found that in most cases, the high spin electronic structures had the most favourable energies and were hence the most stable.

## 1. Introduction

Current environmental concerns are driving significant research into radically new concepts and technologies for sustainable energy production and storage. One such technology is the use of nano-dots and quantum dots. Nano-dots are localised nanometre-scale structures, whereas quantum dots are nanoparticles made from semiconductor materials, all exhibiting localised magnetic or electrical fields at very small scales. These localised properties can be exploited, particularly for use in light-emitting devices [1,2,3], information storage [4,5,6] and energy storage [7,8,9]. Nano-dots can be thought of as small magnets which can switch polarity and this change is exploited exclusively in new hard drives [10], solar cells [11,12], super capacitors [13] and batteries [14,15,16].

Hydrogen (H_2_) is a particularly attractive carbon-free energy source, owing to its high calorific value and its non-polluting character, and it is therefore likely to play a major role in attaining a net zero carbon economy [17]. As such, the production of clean hydrogen has been the topic of extensive research. One way to obtain H_2_ is through the electrocatalytic splitting of water, which is a process that consists of two half reactions, namely the hydrogen evolution reaction (HER) and the oxygen evolution reaction (OER). Due to its energy-expensive nature, research has focused on catalysts for the HER from early transition metals [18], noble metal catalysts [19] and metal–organic framework-based electrocatalysts [20] to non-noble metal-based carbon composites [21]. The latest research [22] is centred on metal nanoclusters and single atom electrocatalysts. In this category, a number of investigations [23,24,25] have shown that partially replacing the platinum (Pt) content with ruthenium (Ru) combined with nano-dot technology results in improved materials properties leading to enhanced hydrogen production [25,26]. Natural Ru has a hexagonal close-packed (*hcp*) crystal structure, but new research [27,28] has shown that face-centred cubic (*fcc*) nanoparticles can be produced as well, which are stable and highly reactive. Small amounts of Ru can increase the hardness of Pt and palladium (Pd) [29], thereby increasing the corrosion resistance in superalloys [30], which are all attractive qualities in the highly corrosive environment of electrochemistry.

In a previous study [31], we focused on the properties of different types of nanoparticles (icosahedral, decahedral, cuboctahedral, cubic and spherical) with a face-centred cubic (*fcc*) packing order, not only for ruthenium (Ru), but also platinum (Pt) and palladium (Pd), where we noted that the properties changed as the particle size increased. The properties of the larger nanoparticles investigated resembled those observed in macro-surfaces, but the smallest nanostructures, especially the Ru nano-dots, behaved differently. For example, in both the Ru nanoparticles and Ru surfaces, no magnetisation was observed as this metal is paramagnetic. However, this was not the case in the Ru nano-dots, and we have therefore decided to investigate these nano-dots in combination with their reactivity towards hydrogen.

In this work, we first discuss the Ru *fcc* nano-dots, both in the icosahedral (13 and 55 atoms) and cubic shape (13 and 63 atoms), presenting their relative energies together with the electron distribution and magnetisation effects. Next, we investigated hydrogen adsorption of the nano-dots, including the effect on magnetisation of the system and the overall system behaviour during adsorption, where we also presented the calculated hydrogen adsorption energies. All these calculations considered both the spin-polarised and non-spin-polarised treatments of the electrons to allow us to investigate both high-spin and low-spin configurations of the electrons, which is relevant as the nano-dots are on the cusp between molecular systems and extended materials.

## 2. Computational Methods

In this study, we utilised and compared two different approaches to study the nano-dots. We used the VASP (Vienna Ab Initio Simulation Package) code, which applies pseudopotentials to describe the banding of electrons, especially where surface and bulk structures are concerned. The other method used here utilises atom-centred basis sets (as implemented in the Gaussian-09 software package) to assign molecular orbitals to molecules or nanostructures. As this work is part of an ongoing programme of research, the methods used are consistent with our previous work [31]. An open-source molecular dynamics engine, OpenMD (v.2.6) [32], was utilised to construct the nanoparticle coordinate files. The Sutton-Chen forcefields [33] were modified for Ru [34,35] in terms of the atomic mass, charges and lattice parameters. The resulting coordination files of the icosahedral and cubic nano-dots were transferred to the VASP and Gaussian software as input to obtain the optimum nanoparticle geometries and energies.

Calculations based on the density functional theory (DFT) were performed on the Ru nano-dots, using VASP version 5.4.1 [36,37,38,39] and the generalized gradient approximation (GGA) in combination with the exchange correlation functional by Perdew, Burke and Ernzerhof (PBE) [40,41]. We applied two long-range dispersion approximations, i.e., the Grimme zero damping DFT-D3 method (DFT-D3) [42] and the DFT-D3(BJ) method by Grimme with Becke–Johnson damping (DFT-D3(BJ)) [43]. We used the projector-augmented wave pseudopotentials (PAW) [44,45] in all calculations to describe the interactions between the core and the valence electrons. The Ru core electrons were defined to contain up to and including 4s orbitals. Because the oxidation states of Ru in compounds can be −2, 0 and +1 up to +8, all the 14 valence electrons in 4p^6^ 4d^7^ 5s^1^ were considered, leading to an increased computational cost. The electron for the H atom was treated as a valence electron. The recommended cut-off of 400 eV for the plane wave basis sets was applied to the valence electrons. In order to break the symmetry, a periodic simulation cell of non-equivalent dimensions, i.e., 12 × 13 × 14 Å, was used to model each nanoparticle, where a vacuum space of at least 10 Å was introduced in all directions to ensure negligible interactions between the nanoparticles in neighbouring cells. We applied a Gaussian smearing [46] of 0.05 eV with a *Γ*-centred Monkhorst-Pack [47] *k*-point mesh of 1 × 1 × 1 for the geometry optimisations and to calculate the energies. We did not use any symmetry constraints for the nanoparticle computations, but we added dipole corrections in all directions to obtain the optimum accuracy. The tetrahedron method with Blöchl corrections [48] was used to obtain the final static simulations to ensure accurate total energies, densities of states and atomic charges. The criterion for the electronic optimisation was set at 10^−5^ eV and the ionic optimisation criterion at 10^−2^ eV·Å^−1^. For the Ru $Fm\overset{-}{3}m$ crystal structure [49], which in our previous benchmarking study [31] had been shown to have a primitive face-centred cubic (*fcc*) cell, we calculated a *fcc* lattice constant of 3.778 Å, which is in excellent agreement with the 3.87 Å observed in experiments [50].

We used a periodic simulation cell of 12 ×13 × 14 Å^3^ to model the isolated H_2_, which ensured negligible interactions with the images in neighbouring cells. Again, we applied the Gaussian smearing [46] of 0.05 eV for the geometry optimisations and energy calculations. We used a *Γ*-centred 1 × 1 × 1 Monkhorst-Pack [47] *k*-point mesh to compute the H_2_ molecule without symmetry constraints, but with the addition of dipole corrections in all directions.

The Gaussian-09 program package [51] was used to perform geometry optimisations of different morphologies, as well as analytical frequency calculations, which were run with the PBE functional [40,41] in combination with the LANL2DZ doublet zeta (ζ) basis set [52], which has an effective core potential for Ru and 6-311+G* on the hydrogen atoms [53,54,55,56,57]. These calculations also included dispersion contributions using the Grimme zero damping DFT-D3 method [42]. Free energies were obtained with thermal corrections and entropies calculated at 298 K, using the analytical frequency functionality as implemented in Gaussian-09 [51]. The zero point corrected total energy ($E_{0}$) is the sum of the zero point vibrational energy (*ZPVE*) and the total electronic energy ($E_{tot}$):(1)$$E_{0}=E_{SCF}+ZPVE$$
where $E_{SCF}$ is the converged self-consistent field energy of the molecular system calculated at the PBE level of theory. *ZPVE* results from the vibrational motion of the molecular systems (even at 0 K) and for a harmonic oscillator model it is calculated as the sum of the contributions from all the vibrational modes of the system.

The average cohesion energy ($E_{coh}$) of the *fcc* Ru nano-dot was defined as follows [58]:(2)$$E_{coh}=\frac{1}{N_{Ru}}\left[E_{Nano}^{DFT}-\left(N_{Ru}\times E_{bulk}^{DFT}\right)\right]$$
where $N_{Ru}$ is the number of atoms in the nano-dot, $E_{Nano}^{DFT}$ is the energy of the Ru nano-dot and $E_{bulk}^{DFT}$ is the energy per atom of the Ru bulk metal. $E_{coh}$ gives the relation between the nano-dot system energy ($E_{Nano}^{DFT}$) vs. the bulk energy ($E_{bulk}^{DFT}$), and thus, as the nano-dot grows in size, $E_{coh}≅0$ eV.

$E_{ads}$ is the average adsorption energy of the H_2_ per molecule adsorbed onto the Ru nano-dot and was calculated as [59,60,61,62]:(3)$$E_{ads}=E_{Ru,r}^{N_{H_{2}}\neq 0}-{(E}_{Ru,r}^{N_{H_{2}}=0}+\frac{1}{2}E_{H_{2}})$$
where $E_{Ru,r}^{N_{H2}\neq 0}$ is the energy of the Ru nano-dot with adsorbed H atom (i.e., half the H_2_ molecule), $E_{Ru,r}^{N_{H2}=0}$ is the energy of the clean Ru nano-dot and $E_{H_{2}}$ is the gas phase energy of the free H_2_ molecule.

Bader analysis [63,64,65,66] was applied to obtain the atomic charges. This method divides space into non-spherical atomic regions, which are enclosed by local minima in the charge density.

We have used the Visualization for Electronic and STructural Analysis (VESTA) v.3.5.5 [67] software (Koichi MOMMA, Fujio IZUMI National Museum of Nature and Science, 4-1-1 Amakubo, Tsukuba, Ibaraki 305-0005, Japan) to produce all the graphics for the surfaces and nanoparticles shown in this work.

## 3. Results and Discussion

### 3.1. Ruthenium Nano-Dot Geometry

In previous work [31], we considered different nanoparticle types and sizes, where we observed that the most stable configuration belonged to the icosahedral nanoparticles of increased sizes, not only for the three metals under investigation (Pd, Pt and Ru), but also using three different computational methods to include dispersion corrections, i.e., DFT-D2, DFT-D3 and DFT-D3(BJ). The surface studies have shown that the surface energies of Ru from all three methods follow the observed trend Ru (111) < Ru (001) < Ru (011). Therefore, in this work we focused on nano-structures that include the (111) Miller index surface. Additionally, to make sure that the observed magnetic moment was not an artifact of the nano-dot shape, we also included cuboctahedral and cubic nano-dots in this study.

Figure 1 depicts the stable Ru *fcc* nano-dots for the icosahedral, cuboctrahedral and cubic morphologies, at each of the three different sizes studied. The icosahedral shape consists of 20 equilateral triangles, all formed from (111) Miller index surfaces. Mackay [68] has shown that the optimum stable configurations for icosahedral structures have a geometrical packing order of G_m_ = 1, 13, 55, 147, 309 and so on, and we have therefore modelled the first three particles in this range. The cuboctahedral nano-dot has a cubic structure that is truncated by 6 square and 8 triangular faces [69] that correspond to Miller index (001) and (111) surfaces, respectively. The packing order is similar to the icosahedral particle with G_m_ = 13, 55, 147. The third nano-dot considered was cubic with 6 square faces, all formed from the (001) Miller index surfaces. Due to the cubic nature, the packing order was slightly different with G_m_ = 13, 63, 171.

As part of our evaluation of the nano-dots for catalytic applications, we investigated the adsorption of a number of molecules onto the nano-dots. During adsorption reactions, electron exchange occurs between the metal and the adsorbate molecule. To facilitate the calculation of this charge transfer, computational settings for VASP included spin-polarised DFT calculations. As the Ru metal is non-magnetic, the spin-polarised DFT calculations aided the electron exchange during catalytic reactions. Figure 2 shows the electron configuration of the Ru d^6^ metal, where the atomic orbitals increased in energy in the order: 1s < 2s < 2p < 3s < 3p < 4s < 3d < 4p < 5s < 4d, with the resulting electron configuration of [Kr] 4d7 5s1.

In non-spin-polarised DFT calculations, the molecular orbitals of the Ru nano-dots were in a low spin configuration, i.e., frontier electrons were distributed over the *π*-orbitals (π_xy_, π_yz_, π_xz_) to give three fully occupied orbitals, exhibiting diamagnetic properties. In spin-polarised DFT calculations, the electrons had a high-spin configuration, i.e., one orbital was fully occupied and four orbitals contained single unpaired electrons, which therefore exhibited paramagnetic properties [70]. This parallel alignment of the electronic spins not only leads to a gain in the exchange energy, but it also causes a loss of kinetic energy [70,71]. The next two sections present and discuss the spin-polarised and non-spin-polarised DFT data.

#### 3.1.1. Spin-Polarised Data

To optimise the geometry of the Ru nano-dots, the DFT-D3 and DFT-D3(BJ) dispersion methods were chosen to account for the effect of non-bonding interactions, with the aim of observing the effect of the different methods on the magnetic moments. Table 1 shows the total energy (*E*_0_ in eV), the average cohesive energy (*E_coh_* in eV) and magnetic moment (*μ_B_*) per atom for the *fcc* Ru icosahedral, cuboctahedral and cubic nano-dots of increasing sizes, obtained with the two dispersion methods. From Table 1, it can be seen that *E_0_* increased as the size of the nano-dot increased, as more atoms (and inherent electrons) were accounted for. This value made it difficult to determine if a nano-dot was stable as there are very few other computational or experimental data available to compare with. The second energy for comparison was the average cohesion energy (*E_coh_*), giving the relationship between the nano-dot energy and the bulk energy. As the nano-dot size increased, *E_coh_* decreased towards $E_{coh}≅0$ eV, which is an indication that the growing nano-dot started exhibiting bulk-like behaviour.

The most interesting result came from the magnetic moments (*μ_B_*) of each of the nano-dots. Again, the trend showed that as the nano-dot increased in size, the magnetic moment per atom was reduced. For the icosahedral nano-dot Ru_13_ with only 13 atoms in the structure, the total magnetic moments (*μ_B_*_,*tot*_) were 20.85 and 12.20 from DFT-D3 and DFT-D3(BJ), respectively. This does not necessarily mean that there were 12–21 unpaired electrons, but rather that a high degree of charge transfer took place, with a subsequent increase in the radical character within the nano-dot. Core electrons from the minority spin channel were promoted to either vacant orbitals located on the surrounding atoms or from atomic orbitals to the bonding molecular orbitals. As these nano-dots were so small, they exhibited behaviour more associated with discrete molecular orbitals of varied orbital energies and less orbital banding, as is the norm in large nanoparticles. Thus, small perturbations of frontier orbitals (e.g., from the binding of a hydrogen atom) can lead to a large degree of splitting in the molecular orbital energies. The splitting of core molecular orbital energies can lead to a large change in the covalent/metallic nature of the Ru–Ru bonds and will induce a large change in the system’s magnetic moment.

To determine any charge distribution in the icosahedral Ru_13_ nano-dot, a Bader charge analysis of the structure was carried out, as shown in Table 2. For all these structures, the first atom was at the core, with 12 surrounding Ru atoms in different orientations. In the icosahedral nano-dot, all 12 surrounding Ru atoms were at a distance of 2.56 Å, indicating that any charge distribution was not related to the inter-atomic distance, since the difference in bond lengths from those calculated with interatomic potentials was less than 3%. The reported Ru–Ru bond lengths have been reported as 2.68 Å [72] in *fcc* bulk structures, 2.26 Å in Ru_2_ clusters modelled with DFT-PBE [73] and 2.41Å [74] with a non-relativistic model potential, showing a clear dependency on the methodology used. The results showed that the core Ru atom donated electron density (Δq = 0.27 or 0.40 e^−^ with D3 or D3(BJ), respectively) to the surrounding atoms, giving rise to the magnetic moment. Similar results were observed in both the cuboctahedral and cubic Ru_13_ nano-dots for the Bader charges and the resulting magnetic moment.

Table 1 shows similar *E*_0_ and *E_coh_* energies for the Ru_13_ cuboctahedral and cubic nano-dots. The biggest variation is in *μ_B_*, where for DFT-D3 and DFT-D3(BJ) we observed a difference of 0.2 and 0.6 per atom, respectively, when compared to the Ru_13_ icosahedral nano-dot. As all 12 surrounding Ru atoms are located at a distance of 2.54 Å, a difference of 0.02 Å (deviating by less than 2% from the interatomic potential model) indicated that this effect was also not an effect of the atomic distance. Again, the Bader charge analysis results in Table 2 showed that the core Ru atom was electron-depleted (Δq = 0.29 or 0.27 e^−^ with D3 or D3(BJ), respectively), but not to such an extent as to explain the difference in *μ_B_*.

#### 3.1.2. Non-Spin-Polarised Data

As the shapes of the Ru_13_ cuboctrahedral and cubic nano-dots are very similar, we carried out the non-spin-polarised calculations only for the icosahedral and cubic nano-dots. Table 3 shows the *E*_0_ and energy difference (∆E_diff_) between the spin-polarised and non-spin-polarised calculations (Table 1) for the *fcc* Ru icosahedral and cubic nano-dots of two sizes, employing both the DFT-D3 and DFT-D3(BJ) methods.

For the icosahedral Ru_13_, we can see from ∆E_diff_ that from both DFT-D3 and DFT-D3(BJ) the high spin state was more favourable. We also included calculations with the spin-polarised contribution, where the initial electron orientation distribution (∆e_↑↓_) configuration was zero, meaning that there was no difference between the alpha- and beta-spin electrons. In the icosahedral Ru_13_ with DFT-D3, the ∆E_diff_ showed that the data resembled a spin-polarised configuration, but for DFT-D3(BJ) they were correlated with a non-spin-polarised configuration. This difference was due to one calculation method forcing a closed shell singlet state more than the other method. In the case of icosahedral Ru_55_, the ∆E_diff_ indicated that the non-spin-polarised configuration was preferred for the DFT-D3 method, while the most stable configuration was spin-polarised with DFT-D3(BJ).

In the case of cubic Ru_13_, similar results to the icosahedral orientation were observed, in that from ∆E_diff_ the high spin state was more favourable for calculations using both DFT-D3 and DFT-D3(BJ). However, in contrast with the icosahedral Ru_13_, here the calculations showed that the spin-polarised calculation with ∆e_↑↓_ = 0 was more stable with DFT-D3 and gave the same results as the normal spin-polarised calculation with DFT-D3(BJ). The cubic Ru_63_ nano-dot showed a more stable configuration with the non-spin-polarised calculation from both DFT-D3 and DFT-D3(BJ) and is thus in a diamagnetic configuration.

As already mentioned, in the non-spin-polarised DFT calculations, the molecular orbitals were in a low spin configuration to give three fully occupied orbitals, with the nano-dot exhibiting diamagnetic properties. In spin-polarised DFT calculations, the electrons have a high-spin configuration, i.e., one orbital is fully occupied and four orbitals contain single unpaired electrons, which therefore exhibits paramagnetic properties [70]. As was the case here, most solid-state systems are non-magnetic, since the gain in exchange energy is outweighed by the loss in kinetic energy, which arises from the delocalization of the valence electrons in a solid [71]. However, even if the solid-state system is intrinsically non-magnetic, the competition between the effects from the exchange and kinetic energies could cause magnetism [70]. The decrease in energy as a result of the exchange effect from an increase in the number of excess parallel spins is accompanied by an increase in energy caused by the electrons moving to higher energy states in the band.

In solid-state chemistry research, most computations employ spin-polarised calculations only, which are normally carried out with a variable magnetic moment. However, this procedure does not guarantee that the lowest energy state is identified. DFT solutions can converge to various local minima, potentially also including metastable states. The final solution often depends on the initial magnetic configuration, since the solution is likely to converge to the nearest local minimum rather than to the global minimum. As the two methods employed here, i.e., spin-polarised and non-spin-polarised computation, gave very different results, it is recommended that especially in small nano-systems, where well-defined surfaces are not observed, e.g., nano-dots rather than nano-particles, both types of calculation should be included.

### 3.2. Hydrogen Adsorption

The reference state for the hydrogen molecule (H_2_) was calculated, producing a formation energy of −4.58 eV from both DFT-D3 and DFT-D3(BJ), which correlates well with experimental data [75] in the gas phase that show the H_2_ formation energy to be −4.48 eV at 298.15 K and 1 atm.

The optimised nano-dots in both the icosahedral (Ru_13_, Ru_55_) and cubic (Ru_13_, Ru_63_) morphologies were used to calculate the hydrogen atom (H) adsorption onto the *fcc* position, which had been shown before [76] to be the most stable configuration. Figure 3 shows the starting structures for the four nano-dots with an adsorbed hydrogen, as well as the resulting optimised structures, viewed from different angles.

The calculations for all the nano-dots included both the spin-polarised and non-spin-polarised settings, and for the icosahedral and cubic Ru_13_ nano-dots they also included calculations with the initial magnetic moment set to one (*μ_i_* = +1) to account for the addition of H, in addition to the spin-polarised calculations with the ∆e_↑↓_ configurations set both to zero and to one. Table 4 lists the resulting data from the DFT-D3 and DFT-D3(BJ) methods.

The spin-polarised and non-spin-polarised calculations for the icosahedral Ru_13_ nano-dot again indicated that the spin-polarised calculation led to the more stable structure, resulting in a magnetic moment of ~10 *μ*_*B*,*tot*_ with both DFT-D3 and DFT-D3(BJ). Unexpectedly, the nano-dot changed configuration, as shown in Figure 3, still with a central Ru atom, but now the 12 surrounding Ru form two 5-membered rings and one 4-membered ring, and the H is adsorbed onto a bridge position. This structure was not in the *fcc* packing order, but rather resembled an *hcp* distribution. Even though the adsorption data fit within the H-adsorption data for other metals, it is surprising that this *hcp* orientation was only observed after adsorption occurred. This behaviour strongly indicated a change in the Ru–Ru bonding that was indicative of more covalent bonding perpendicular to the hydrogen adsorption site, with electrons promoted from the central Ru $\pi_{xy}$, $\pi_{yz}$, $\pi_{xz}$ orbitals into the unoccupied molecular bonding orbitals $\sigma_{{x^{2}-y}^{2}}$ and $\sigma_{z^{2}}$.

In the case of the cubic Ru_13_ nano-dot, the data from both DFT-D3 and DFT-D3(BJ) suggested that the non-spin-polarised state was more stable. Again, the spin-polarised calculations led to a resulting magnetic moment of ~6–10 *μ*_*B*,*tot*_. Similar to the icosahedral Ru_13_ nano-dot geometry, the cubic structure was reordered from the *fcc* to *hcp* packing order, forming 4- and 5-membered rings around a central Ru atom. Again, the adsorbed H atom was sited in the bridge position.

For the icosahedral Ru_55_, the data deviated in that the spin-polarised calculation for the adsorbed H system led to a more stable structure than that resulting from the non-spin-polarised calculation. As expected, the spin-polarised calculation led to a magnetic moment of ~15 *μ*_*B*,*tot*_ and no restructuring of the *fcc* packing order was observed, whereas the H atom was adsorbed in the *fcc* hollow.

In the case of the cubic Ru_63_ nano-dot, the data correlated with the clean nano-dot calculations, whereby the non-spin-polarised calculation led to a more stable structure when H was adsorbed. However, the resulting *μ*_*B*,*tot*_ was much lower than that observed for the clean cubic Ru_63_ nano-dot. Although this nano-dot still had the *fcc* packing order, changes in the structure were observed with the morphology tending toward a cuboctrahedral configuration, as shown in Figure 1, indicating that the formation of the (111) Miller planes stabilised the structure. Similar to the icosahedral Ru_55_ nano-dot, the adsorbed H atom was in the *fcc* hollow.

### 3.3. Gaussian Correlation

As the nano-dots are very small in size and could be considered not to be proper nanoparticles, or yet to contain well-defined surfaces or bulk, we were interested in investigating how the atom-centred basis set calculations would describe H adsorption onto nano-dots, using software optimised for calculating molecular orbitals. Table 5 tabulates the total energy (*E*_0_), adsorption energy (E_ads_) and the energy difference (∆E_diff_) for both low spin and high spin calculations, utilising PBE/LANL2DZ(6-311+G*), as implemented in Gaussian-09. Again, we first calculated our reference state for H_2_ in the gas phase, where the resulting formation energy of 4.19 eV was slightly under-estimated, compared to the experimental data at a higher temperature and pressure (−4.48 eV at 298.15 K and 1 atm [75]).

The same starting structures for the icosahedral and cubic Ru_13_ nano-dots (Figure 1) and their respective H-adsorbed counterparts (Figure 3) were used. Frequency calculations were carried out for both high spin and low spin hydrogen adsorption to obtain the zero-point vibrational energy values (ZPVE). Comparing the ∆E_diff_, we can see that in the case of the icosahedral Ru_13_ nano-dot, the high spin calculation gave a more stable structure and the same is true for H adsorption in the high spin state. However, for the cubic Ru_13_ nano-dot, the low spin state was more stable. Similar to what was seen in the VASP calculations (Figure 3), restructuring of the Ru atoms occurred, whereby one central Ru atom was surrounded by 4- and 5-member rings. Once H was adsorbed, it is evident that the low spin state remained the more stable configuration as E_ads_ was negative, a general indication of exothermicity. Overall, these data correlate with the VASP calculations, which provides us with confidence that the calculations reflect the properties of the materials.

## 4. Conclusions

Calculations based on the density functional theory (DFT) have been employed to gain detailed insight into the behaviour of different types and sizes of ruthenium nano-dots and how spin-polarised and non-spin-polarised calculations affect the magnetic moments and total energy. Two dispersion methods were used, i.e., DFT-D3 and DFT-D3(BJ).

For the Ru_13_ nano-dots of different morphologies (icosahedral and cubic), it was seen that the spin-polarised calculations led to the more stable structure. However, for the icosahedral Ru_55_ and cubic Ru_63_ this was not the case. Hydrogen adsorption showed that the initial magnetic moment was reduced, and that the spin-polarised calculations led to the more stable structures. Overall, in all cases the DFT-D3 method overestimated the magnetic moment compared to the DFT-D3(BJ) method. However, we observed that restructuring of the cubic Ru_13_ occurred not only for the pseudopotential-generated data (VASP), but also in the atom-based basis set data (Gaussian).

## Figures and Tables

**Figure 1 nanomaterials-13-01118-f001:**
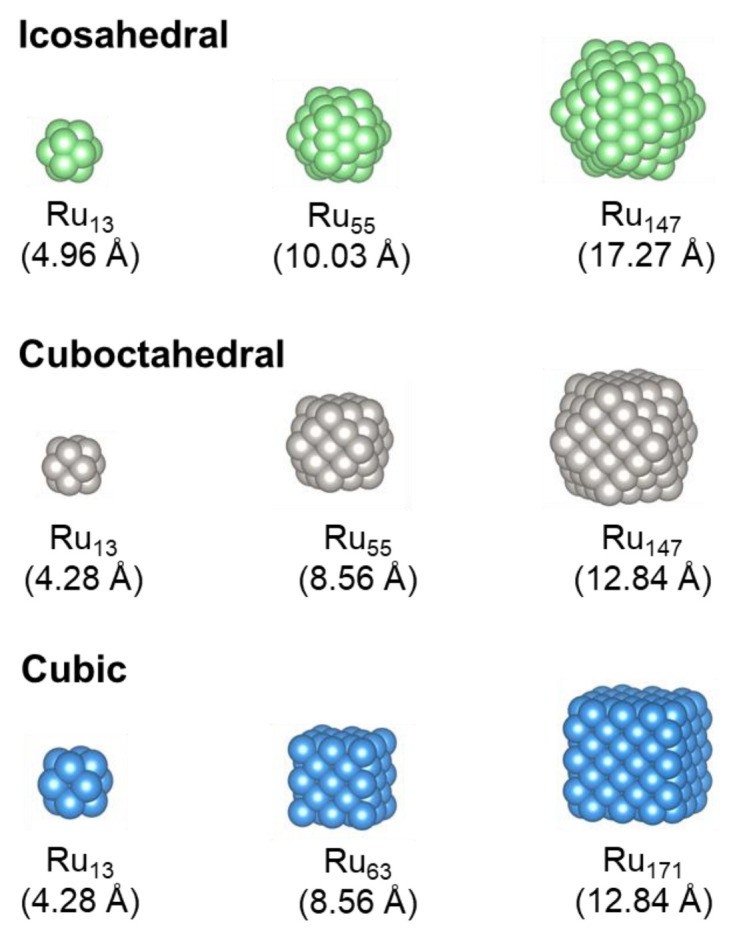
Stable Ru *fcc* nano-dots for the icosahedral, cuboctrahedral and cubic shapes, at each of the three sizes studied; calculated with the interatomic potentials for Ru [34,35] utilised in OpenMD [32].

**Figure 2 nanomaterials-13-01118-f002:**
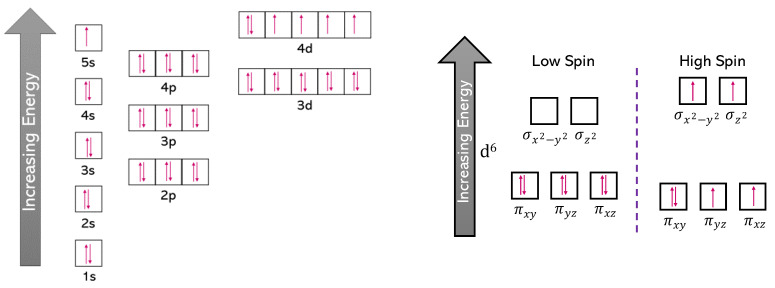
A schematic representation of Ru *d*^6^ atomic orbitals (**left**) as well as the potential molecular orbitals occupations (**right**) are both shown.

**Figure 3 nanomaterials-13-01118-f003:**
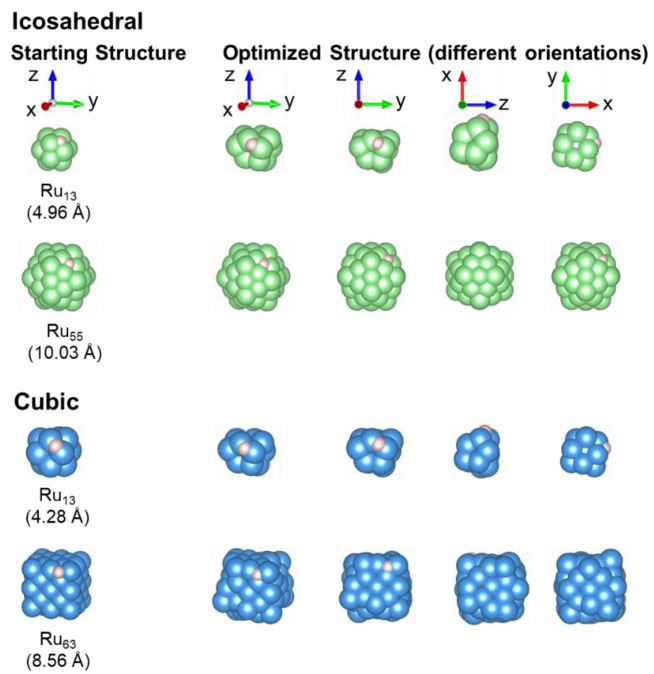
Hydrogen adsorption onto the icosahedral and cubic Ru *fcc* nano-dots for two different sizes, showing initial and optimised structures.

**Table 1 nanomaterials-13-01118-t001:** Total energy (*E*_0_), cohesive energy (*E_coh_*) and magnetic moment (*μ_B_*) for the *fcc* Ru icosahedral, cuboctahedral and cubic nano-dots of increased sizes with two modelling methods, DFT-D3 and DFT-D3(BJ), respectively.

	DFT-D3			DFT-D3(BJ)		
	*E*_0_ (eV)	*E_coh_* (eV)	*μ* * _B_ *	*E*_0_ (eV)	*E_coh_* (eV)	*μ* * _B_ *
Icosahedral						
Ru_13_ (4.96 Å)	−87.670	2.893	1.60	−88.307	2.948	0.94
Ru_55_ (10.03 Å)	−435.576	1.718	0.28	−439.517	1.750	0.28
Ru_147_ (17.27 Å)	−1241.226	1.193	0.00	−1251.887	1.225	0.00
Cuboctahedral						
Ru_13_ (4.28 Å)	−86.745	2.964	1.39	−89.956	2.821	0.31
Ru_55_ (8.56 Å)	−434.176	1.743	0.00	−438.140	1.775	0.00
Ru_147_ (12.84 Å)	−1227.905	1.284	0.00	−1238.298	1.317	0.00
Cubic						
Ru_13_ (4.28 Å)	−86.745	2.964	1.39	−90.268	2.797	0.32
Ru_63_ (8.56 Å)	−491.376	1.837	0.15	−502.370	1.767	0.03
Ru_171_ (12.84 Å)	−1423.041	1.315	0.05	−1435.418	1.347	0.04

**Table 2 nanomaterials-13-01118-t002:** Results of the Bader charge analysis (∆q/e^−^) of the smallest system (Ru_13_) for the icosahedral, cuboctahedral/cubic nano-dots, from DFT-D3 and DFT-D3(BJ).

	Icosahedral		Cuboctahedral/Cubic	
Atom #	DFT-D3	DFT-D3(BJ)	DFT-D3	DFT-D3(BJ)
1 (core)	0.269	0.404	0.290	0.267
2	−0.055	−0.011	0.040	−0.014
3	0.010	−0.066	−0.152	−0.030
4	−0.055	−0.060	0.102	0.008
5	0.010	−0.017	−0.023	−0.047
6	0.010	0.013	0.102	−0.019
7	−0.055	−0.064	−0.151	−0.002
8	0.010	−0.061	−0.089	−0.084
9	0.010	0.012	−0.087	−0.036
10	−0.055	−0.072	−0.023	−0.065
11	−0.055	−0.063	0.040	−0.018
12	0.010	0.002	0.102	−0.051
13	−0.055	−0.017	−0.152	0.089

**Table 3 nanomaterials-13-01118-t003:** Total energy (*E*_0_) and energy difference (∆E_diff_) for the *fcc* Ru icosahedral and cubic nano-dots of two sizes with two modelling methods to include the non-spin-polarised contribution and initial electron orientation distribution (∆e_↑↓_) configurations for both DFT-D3 and DFT-D3(BJ).

	DFT-D3		DFT-D3(BJ)	
	*E*_0_ (eV)	∆E_diff_ (eV)	*E*_0_ (eV)	∆E_diff_ (eV)
Icosahedral				
Ru_13_ (4.96 Å)	−86.38	1.29	−87.95	0.35
∆e_↑↓_ = 0	−87.73	−0.06	−87.94	0.36
Ru_55_ (10.03 Å)	−435.62	−0.04	−439.09	0.42
Cubic				
Ru_13_ (4.28 Å)	−86.64	0.10	−90.16	0.11
∆e_↑↓_ = 0	−88.25	−1.50	−90.27	0.00
Ru_63_ (8.56 Å)	−491.46	−0.09	−503.33	−0.96

**Table 4 nanomaterials-13-01118-t004:** System energy (*E*_0_), adsorption energy (E_ads_) and total resulting magnetic moment (*μ_B_*_,*tot*_) for the *fcc* Ru icosahedral and cubic nano-dots of two sizes, with H-adsorption onto the *fcc* hollow site to include the spin-polarised (SP) and non-spin-polarised (NSP) contributions, different initial magnetic moments (*μ_i_*) and initial electron orientation distribution (∆e_↑↓_) configurations, obtained with DFT-D3 and DFT-D3(BJ).

		DFT-D3			DFT-D3(BJ)		
		*E*_0_ (eV)	E_ads_ (eV)	*μ_B_* _,*tot*_	*E*_0_ (eV)	E_ads_ (eV)	*μ_B_* _,*tot*_
Icosahedral							
Ru_13_ + 1H	SP	−91.90	−0.58	10.32	−92.89	−1.17	10.69
	NSP	−91.21	−0.13	0.00	−92.32	−0.61	0.00
	*μ_i_* = +1	−91.56	−0.48	9.81	−92.01	−0.30	0.05
	∆e_↑↓_ = 0	−91.20	−0.13	0.00	−92.02	−0.31	0.00
	∆e_↑↓_ = 1	−91.22	−0.15	1.00	−92.05	−0.34	1.00
Ru_55_ + 1H	SP	−439.80	−0.82	15.29	−443.78	−0.86	15.33
	NSP	−439.37	−0.39	0.00	−443.35	−0.42	0.00
Cubic							
Ru_13_ + 1H	SP	−91.36	−1.21	10.79	−94.29	-0.61	6.01
	NSP	−92.75	−2.60	0.00	−94.51	−0.84	0.00
	∆e_↑↓_ = 0	−92.87	−2.72	0.00	−93.77	−0.10	0.00
	∆e_↑↓_ = 1	−92.13	−1.98	1.00	−93.86	−0.18	1.00
Ru_63_ + 1H	SP	−502.22	−7.44	1.47	−510.54	−4.77	0.00
	NSP	−503.38	−8.60	0.00	−510.60	−4.83	0.00

**Table 5 nanomaterials-13-01118-t005:** System energy (*E*_0_), adsorption energy (E_ads_) and energy difference (∆E_diff_) for the *fcc* Ru icosahedral and cubic nano-dots of two sizes with H-adsorption onto the *fcc* hollow site to include the high and low spin contributions with Gaussian-09 (with the PBE functional [40,41] in combination with the LANL2DZ doublet zeta (ζ) basis set [52]).

	Low Spin		High Spin		
	*E*_0_ (eV)	E_ads_ (eV)	*E*_0_ (eV)	E_ads_ (eV)	∆E_diff_ (eV)
1H	−13.60				
H_2_	−31.39				−4.19
Icosahedral	−33,219.29		−33,219.97		−0.68
Ru_13_ + ½ H_2_	−33,235.60	−0.62	−33,236.16	−0.49	−0.55
Cubic	−33,220.46		−33,219.51		0.95
Ru_13_ + ½ H_2_	−33,236.82	−0.66	−33,234.88	0.33	−1.94

## Data Availability

All data created during this research are openly available from Cardiff University’s Research Portal: M.J. Ungerer and N.H. de Leeuw (2022). “A DFT study on Ruthenium *fcc* Nano-dots: Size Dependent Induced Magnetic Moment”, Cardiff University’s Research Portal, V. 1, Dataset. http://doi.org/10.17035/d.2023.0238557525 (accessed on 24 January 2023).

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
