# Peer review of "A DFT Study of Ruthenium fcc Nano-Dots: Size-Dependent Induced Magnetic Moments"

_nanomaterials, 2023, doi:10.3390/nano13061118_

Round 1

Reviewer 1 Report

In this manuscript, density functional theory (DFT) calculations were carried out with long-range dispersion corrections [DFT-D3 and DFT-D3-(BJ)] to systematically investigate the magnetic moments of two different morphologies and various sizes of Ru nano-dots in the fcc phase. In addition, atom-centred DFT calculations were carried out on the smallest nano-dots to establish accurate spin-splitting energetics to confirm the results obtained by the plane-wave DFT methodologies. Here are some comments for this manuscript:

1. The email address belongs to MJU, please label the * for MJU.

2. Line 96. Non-equivalent dimensions were used in this manuscript. Are there any specific reasons that the calculations must use non-equivalent dimensions?

3. Line 159. What is Mackay? There is no reference [86] in this manuscript.

4. Line 227-228. How to obtain the distance of 2.56 Å? The size of icosahedral Ru13 is 4.96 Å (Figure 1) and 2 * 2.56 Å > 4.96 Å. Why the distance of 2.56 Å can obtain the conclusion that any charge distribution is not related to inter-atomic distance?

5. Line 229. Why the number of electrons that the core Ru donates are calculated using Δq = 0.27 – 0.40 e−? Δq/e− in Table 2 already indicates the electron donation or receiving for Ru atom.

6. Line 237. How to obtain the difference of 0.2 and 0.3 per atom for DFT-D3 and DFT-D3(BJ), respectively?

7. Line 239. Same as comment 4. How to obtain the distance of 2.54 Å?

8. Line 241. Same as comment 5. Why the number of electrons that the core Ru donates are calculated using Δq = 0.27 – 0.29 e−?

9. Line 368. Are ZPVE calculated only for H adsorption? Do the total energy and system energy E0 in table 1, 3, 4 and 5 also include ZPVE?

10. Line 369-370. The high spin calculation gave a more stable structure and the same is true for H adsorption in the high spin state. The adsorption energy (Eads) for high spin is -0.49 eV, which is higher than that for low spin (-0.62 eV). The adsorption of H in the low spin state should be more stable.

Reviewer 2 Report

This manuscript demonstrates a systematic study of Ru nanoparticles. Both PBC and non-PBC calculations were performed using VASP and Gaussian09, respectively. The PBE exchange-correlation functional, basis set, and PAW were appropriate for studying the transition metal clusters. Both PAW and AO basis sets showed similar results. The conclusion is therefore solid at the DFT level. I suggest this paper be published after few revisions.

1.      Some formatting issues. Page 3: Please align the equations correctly. Equation font sizes: please make the font size consistent with the main body.

2.      If the nanoparticle follows the equilibrium growth rules, such as Wulff construction, the structures would demonstrate a surface energy dependence. Several published databases describe the surface energy and shape of the metal clusters, for instance, crystalium.materialsvirtuallab.org. How are your results compared with Wulff construction results?

3.      DFT-D3 correction is intended to correct the dispersion of vdW interactions. Does your system benefit from including D3 correction?

The above issues will not greatly impact the major conclusion. Therefore, the paper can be accepted after a minor revision.
